# Mesh-RFT: Enhancing Mesh Generation via Fine-Grained Reinforcement Fine-Tuning

**Jian Liu**[1,2*]  **Jing Xu**[2*]  **Song Guo**[1†]  **Jing Li**[2,3]  **Jingfeng Guo**[2,4]  **Jiaao Yu**[2]
**Haohan Weng**[2,4]  **Biwen Lei**[2]  **Xianghui Yang**[2]  **Zhuo Chen**[2]  **Fangqi Zhu**[1]
**Tao Han**[1]  **Chunchao Guo**[2†]

[1] Hong Kong University of Science and Technology    [2] Tencent Hunyuan
[3] University of Science and Technology of China    [4] South China University of Technology

https://hitcslj.github.io/mesh-rft/

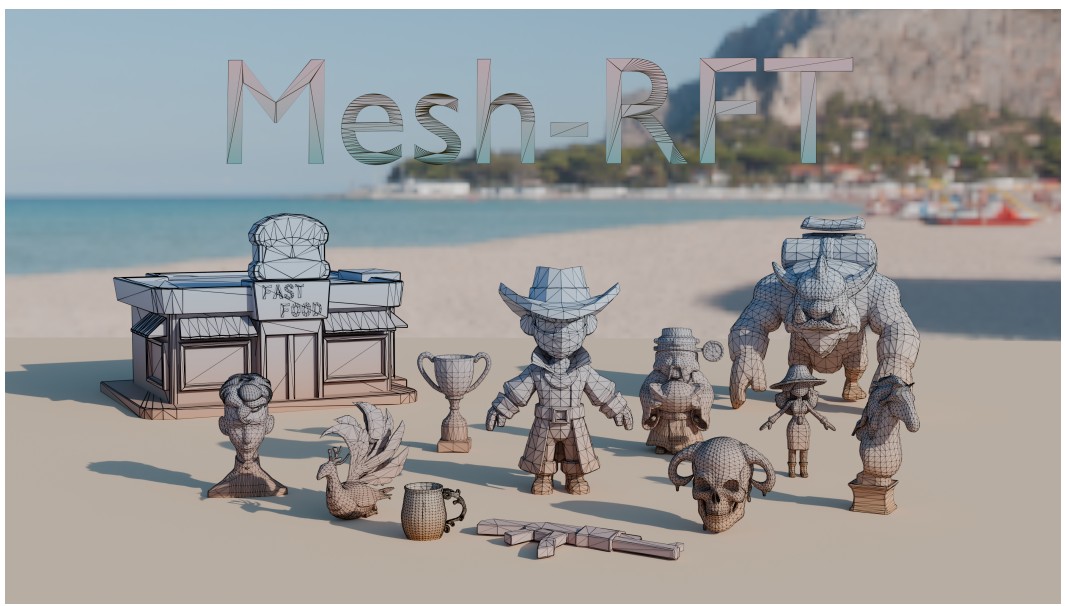

Figure 1: **Representative High-Fidelity Mesh Generation by Mesh-RFT.** Gallery of meshes generated from point clouds, demonstrating intricate geometric detail and artist-like aesthetic quality.

## Abstract

Existing pretrained models for 3D mesh generation often suffer from data biases and produce low-quality results, while global reinforcement learning (RL) methods rely on object-level rewards that struggle to capture local structure details. To address these challenges, we present **Mesh-RFT**, a novel fine-grained reinforcement fine-tuning framework that employs Masked Direct Preference Optimization (M-DPO) to enable localized refinement via quality-aware face masking. To facilitate efficient quality evaluation, we introduce an objective topology-aware scoring system to evaluate geometric integrity and topological regularity at both object and face levels through two metrics: Boundary Edge Ratio (BER) and Topology Score (TS). By integrating these metrics into a fine-grained RL strategy, Mesh-RFT becomes the first method to optimize mesh quality at the granularity of individual

---

[*]Equal Contribution.
[†]Corresponding Author.

39th Conference on Neural Information Processing Systems (NeurIPS 2025).

faces, resolving localized errors while preserving global coherence. Experiment results show that our M-DPO approach reduces Hausdorff Distance (HD) by 24.6% and improves Topology Score (TS) by 3.8% over pre-trained models, while outperforming global DPO methods with a 17.4% HD reduction and 4.9% TS gain. These results demonstrate Mesh-RFT's ability to improve geometric integrity and topological regularity, achieving new state-of-the-art performance in production-ready mesh generation.

# 1   Introduction

3D polygonal meshes serve as the foundational representation for digital assets in industries such as gaming, film, and product design. Despite their ubiquity, high-quality, topologically optimized meshes—essential for downstream tasks like editing, rigging, and animation—are still predominantly handcrafted by skilled artists. Recent advances in generative models have enabled automated mesh synthesis, significantly reducing the time and expertise required to produce production-ready 3D assets. This democratization of mesh generation broadens access to 3D content creation, empowering non-experts to produce geometrically precise and artistically viable models for applications ranging from immersive media to industrial design.

Existing 3D generative models often use intermediate representations like voxels [1, 2], point clouds [3, 4, 5], latent space [6, 7] or implicit fields [8, 9]. While these avoid direct mesh generation complexities, post-processing (e.g., Marching Cubes [10]) often introduces topological issues and smoothing. Native mesh generation [11] is more direct, with recent work using autoregressive models and neural compression (e.g., VQ-VAE [12, 13, 14]) or geometric serialization tokenizers (e.g., [15, 16, 17, 18, 19]) for sequence-based generation. However, long sequences for high-resolution meshes can cause structural ambiguities and hallucinations (inconsistent edges, non-manifold vertices, distortions, holes), deviating from geometric constraints or artistic intent, ultimately leading to results that may not align with human aesthetic preferences or intended design. Though truncated training [20] helps, autoregressive methods still lack stable generation and high fidelity.

Recently, reinforcement learning [21, 22] has emerged as a compelling approach for aligning mesh generation more closely with human preferences. For example, DeepMesh [23] leverages Direct Preference Optimization (DPO) [24], a simple yet effective preference alignment technique that has also found utility in various other domains [25, 26, 27]. Nevertheless, directly applying reinforcement fine-tuning to mesh generation using this method encounters two primary challenges. Firstly, objectively quantifying mesh quality is difficult. DeepMesh relies on manual annotation of preference pairs, which is expensive, time-consuming, introduces subjective bias, and limits the training data to only 5,000 samples, hindering generalization. Secondly, its use of global reward signals fails to capture the local topological variations inherent in 3D meshes. As illustrated in Figure 2, high-quality and low-quality structures often coexist within a single mesh, leading to training noise due to this mismatch in supervision.

To overcome these limitations, we introduce **Mesh-RFT**, a novel framework that combines **Masked Direct Preference Optimization (M-DPO)** with fine-grained mesh quality evaluation for both global and localized refinement. Unlike prior work using subjective global rewards as supervision signals [23], we employ a topology-aware scoring system with automated metrics-Boundary Edge Ratio (BER) and Topology Score (TS)-to objectively evaluate mesh quality at both object and face levels, circumventing the laborious manual annotation efforts. Mesh-RFT further employs a localized optimization mechanism utilizing M-DPO and quality-aware masks to specifically refine defective regions, thereby addressing the coarse supervision

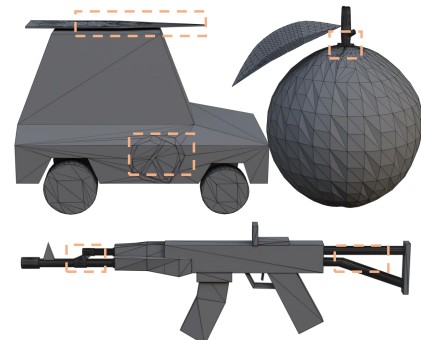

Figure 2: High-quality, artist-like structures often co-exist with messy, low-quality regions within the same mesh.

of global rewards. Extensive experiments across diverse meshes demonstrate Mesh-RFT's superior performance, achieving significant improvements over both the pretrain baseline (24.6% HD

reduction, 3.8% TS improvement) and global DPO (17.4% HD reduction, 4.9% TS improvement), establishing a new benchmark for accuracy and fidelity in generative mesh modeling.

In summary, our contributions are as follows:

- We introduce the first **fine-grained** reinforcement fine-tuning framework, that integrates Masked Direct Preference Optimization (M-DPO) with fine-grained mesh quality evaluation.

- We devise an objective topology-aware scoring system for evaluating mesh quality, eliminating dependency on manual annotation and addressing subjectivity and scalability limitations.

- We propose a novel localized alignment mechanism that optimizes deficient regions geometrically and topologically via quality-aware masks, bridging the gap between global and local supervision.

- Experiments demonstrate that our method achieves state-of-the-art performance in high-fidelity 3D mesh generation.

## 2   Related work

### 2.1   3D Generation via Alternative Representations

Many 3D generative models avoid direct mesh modeling by using intermediate representations like voxels, point clouds, or implicit fields. Early voxel methods [1, 2] using grids faced memory issues. Point cloud methods [3, 4, 28, 29] with networks like PointNet [5, 30] struggle with consistency and detail. Implicit fields, especially neural fields [8, 9, 31, 32], offer efficient representations. These include score distillation with 2D diffusion models [33, 34, 35, 36, 37, 38] and 3D Transformer models like LRM [39, 40, 41, 42, 43], alongside recent latent diffusion methods [44, 45, 46, 47, 48, 49, 50, 51] that have demonstrated good scalability and performance. However, these approaches often rely on post-processing via Marching Cubes [10], which can cause topological issues, smoothing, and artifacts.

### 2.2   Native Mesh Generation

While neural shape representations such as implicit fields have been extensively studied, native mesh generation is an emerging area of research. Early approaches leveraging surface patches [52] or mesh graphs [53] often suffered from quality limitations. Diffusion-based methods [54, 55] have seen limited exploration in this domain, potentially due to inherent difficulties in directly processing meshes. PolyGen [11] demonstrated promise by autoregressively generating mesh vertices and faces. MeshGPT [12] encoded meshes into quantized tokens using VQ-VAE [56] for autoregressive generation. Subsequently, MeshXL [15] proposed a one-stage autoregressive model operating on coordinate-level mesh sequences. Various tokenization techniques [16, 17, 19, 57] and efficient training strategies [20, 58] have been explored to address the challenges of long sequences in high-resolution generation; however, achieving stable and high-fidelity results remains a significant hurdle.

### 2.3   Reinforcement Learning for Mesh Generation

Reinforcement Learning (RL) [59] has gained traction for 3D generation [60] using human feedback. Reinforcement Learning from Human Feedback (RLHF) aligns models with preferences by training a reward model, then fine-tuning with RL. However, RLHF is costly and unstable for 3D tasks. Direct Preference Optimization (DPO) [24] offers a more efficient, stable alternative by removing the reward model. Despite success in language and image domains [25, 26], DPO's application to 3D meshes is limited. Closely related, DeepMesh [23] uses global rewards for alignment but struggles with 3D mesh heterogeneity, over-optimizing some regions and under-optimizing others. Thus, RL methods addressing local mesh structures are crucial for better 3D mesh quality and consistency.

## 3   Method

This section details the Mesh-RFT framework. As illustrated in Figure 3, our pipeline consists of three stages: First, supervised pretraining is performed by feeding point clouds and ground truth

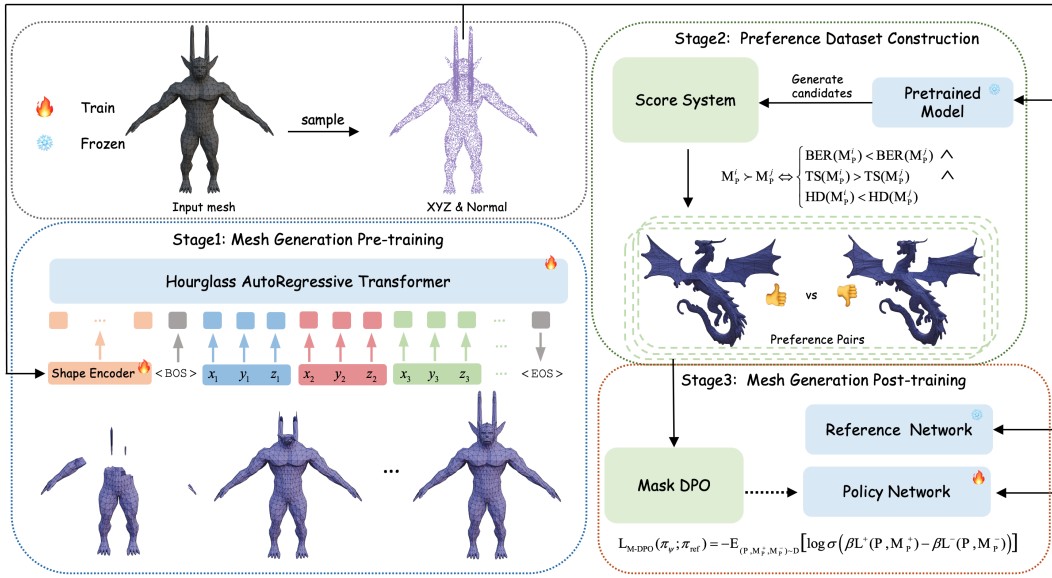

Figure 3: **Mesh-RFT Framework Overview.** The pipeline comprises three stages: **1) Mesh Generation Pre-training** using an Hourglass AutoRegressive Transformer and a Shape Encoder; **2) Preference Dataset Construction** where a pretrained model generates candidate meshes, and a topology-aware score system establishes preference pairs; and **3) Mesh Generation Post-training** which employs Mask DPO with reference and policy networks for subsequent refinement.

mesh sequences into the model. Second, the pretrained model generates candidates, and a topology-aware score system builds a preference dataset. Third, topology-aware Masked Direct Preference Optimization is applied to post-train the model using this preference dataset to refine its performance.

## 3.1 Mesh Generation Pre-training

Firstly, we discuss mesh tokenization. Prior works [16, 17, 19] compress mesh sequences to manage sequence growth with increasing faces, but such techniques embed excessive geometric information per token, causing cascading face errors when a single token is incorrect (e.g., BPT [19] often introduces patch-level holes). To avoid these issues, we adopt the uncompressed mesh sequence method introduced from MeshXL [15]. Specifically, for a given mesh $\mathcal{M}$, we first quantize the vertex coordinates of each face, and then flatten them in $XYZ$ order to construct a complete token sequence.

**Model Architecture.** To better capture the structure of the mesh, rather than framing mesh generation as a generic sequence task, we utilize Hourglass Transformer architecture [20, 61]. Our model processes inputs hierarchically and incorporates two shorten and two upsample operations. The shorten operations reduce the token sequence length using techniques such as linear or attention-based pooling, while the upsample operations expand the sequence back to its original length through linear or attention-based methods. This design enables the model to efficiently capture both high-level patterns and fine-grained details. In point-cloud conditioned mesh generation, achieving fine-grained and complex structures requires not only a powerful decoder but also high-quality point cloud features. To this end, we adopt the point cloud encoder pretrained in Hunyuan3D 2.0 [48] to do this. These features are injected into our autoregressive decoder as keys and values via cross-attention [62].

**Truncated Training and Sliding-Window Inference.** To reduce memory and computational costs, we employ truncated training with fixed-length segments. This approach involves extracting smaller, fixed-length segments from the mesh sequence for training, rather than using the entire sequence. When a segment does not contain the start-of-sequence (SOS) token, we pad a small prefix portion to avoid misleading the model. During inference, we use a sliding window approach to enhance both speed and generation quality. The sliding process begins once $40\%$ of the training window size is covered, and only the most recent $30\%$ of tokens are retained. This method reduces computational load by focusing on the most relevant tokens, as distant tokens typically have less influence on each

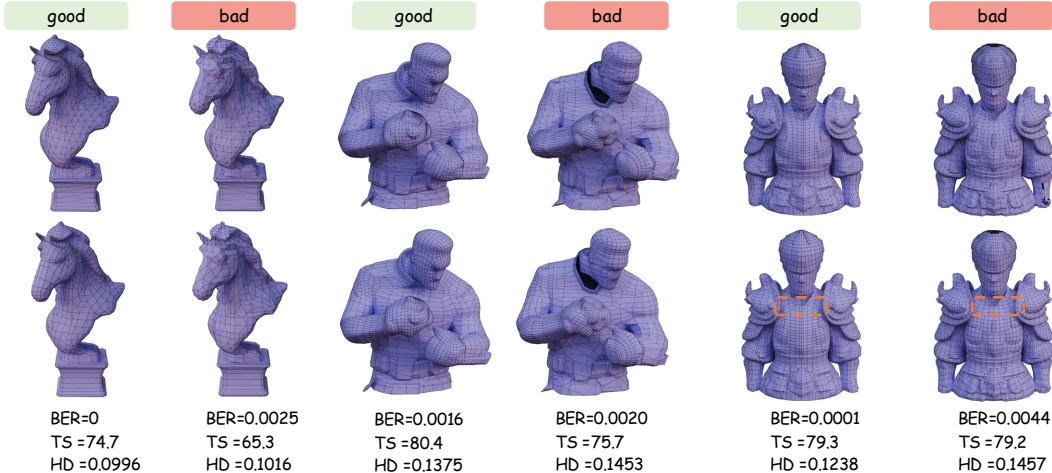

| good | bad | good | bad | good | bad |
|------|-----|------|-----|------|-----|
| BER=0 | BER=0.0025 | BER=0.0016 | BER=0.0020 | BER=0.0001 | BER=0.0044 |
| TS =74.7 | TS =65.3 | TS =80.4 | TS =75.7 | TS =79.3 | TS =79.2 |
| HD =0.0996 | HD =0.1016 | HD =0.1375 | HD =0.1453 | HD =0.1238 | HD =0.1457 |

Figure 4: **Examples of collected preference pairs.** Meshes are annotated as preferred using our scoring system. For certain pairs, the selected "good" meshes may exhibit inferior local performance in specific regions compared to the rejected "bad" meshes.

other. Additionally, it helps mitigate high perplexity at the tail of each window, leading to more accurate and efficient generation.

### 3.2 Preference Dataset Construction

We establish a systematic pipeline for constructing the preference dataset, which is used for RLHF fine-tuning in the second stage. This pipeline consists of three key components: candidate generation, multi-metric evaluation, and preference ranking. The process is described as follows.

**Candidate Generation.** For each input point cloud $\mathcal{P}$, we generate eight candidate meshes $\left\{\mathcal{M}_{\mathcal{P}}^1, \mathcal{M}_{\mathcal{P}}^2, \cdots, \mathcal{M}_{\mathcal{P}}^8\right\}$ using the pre-trained model $G_\theta^{pre}$.

**Multi-Metric Evaluation.** We evaluate each candidate mesh using a comprehensive set of criteria to assess both geometric consistency and topological quality. In addition to measuring the geometric alignment with the input data, we introduce two topology-oriented metrics that specifically aim to capture the structural integrity and coherence of the generated meshes. These three metrics are: Boundary Edge Ratio (BER) and Topology Score (TS) for evaluating topology, and Hausdorff Distance (HD) for evaluating geometric consistency.

- **Boundary Edge Ratio (BER)**: This metric, defined as $BER(\mathcal{M}) = \frac{E_{\partial\mathcal{M}}}{E_\mathcal{M}}$, quantifies the integrity of the mesh by calculating the proportion of its boundary edges ($E_{\partial\mathcal{M}}$) to the total number of edges ($E_\mathcal{M}$). Boundary edges are those connected to only one face, and a high BER value (typically above 0.002 in our dataset, which consists mostly of closed meshes) suggests potential issues like surface discontinuities, holes, or mesh damage. Ideally, a closed, manifold mesh should have a BER of 0.

- **Topology Score (TS)**: The Topology Score, $TS(\mathcal{M}) = \sum_{i=1}^{4} w_i s_i(\mathcal{Q}(\mathcal{M}))$, assesses the structural quality of a mesh $\mathcal{M}$ by analyzing a derived quadrilateral mesh $\mathcal{Q}(\mathcal{M})$, obtained through standard triangle-to-quad merging. The score is a weighted sum of four sub-metrics: Quad Ratio ($w_1 = 0.4$), which measures the efficiency of the conversion; Angle Quality ($w_2 = 0.2$), quantifying the deviation of quadrilateral angles from $90°$; Aspect Ratio ($w_3 = 0.3$), evaluating the regularity of quadrilateral shapes; and Adjacent Consistency ($w_4 = 0.1$), encouraging uniform aspect ratios between neighboring quadrilaterals. This quadrilateral-based evaluation is used because quad meshes are preferred in industrial applications, making the quality of the quadrangulation a practical indicator of the topological soundness of the original triangular mesh. Further details are in the supplementary material A.3.

- **Hausdorff Distance (HD)**: This standard metric measures the maximum distance from a point in one set to the closest point in the other set. Here, it quantifies the geometric alignment between the

reconstructed mesh $\mathcal{M}_{\mathcal{P}}^i$ and the input point cloud $\mathcal{P}$ by measuring the maximum distance between their respective point samples. A lower HD value indicates a better geometric reconstruction.

**Preference Ranking.** To construct the preference dataset, we generate pairwise comparisons through exhaustive combinations of the eight candidate meshes for each input point cloud $\mathcal{P}$, resulting in a total of $\binom{8}{2} = 28$ pairs. For each pair $(\mathcal{M}_{\mathcal{P}}^i, \mathcal{M}_{\mathcal{P}}^j)$, we define a preference relation $\mathcal{M}_{\mathcal{P}}^i \succ \mathcal{M}_{\mathcal{P}}^j$ if and only if $\mathcal{M}_{\mathcal{P}}^i$ outperforms $\mathcal{M}_{\mathcal{P}}^j$ across all three evaluation metrics:

$$
\mathcal{M}_{\mathcal{P}}^i \succ \mathcal{M}_{\mathcal{P}}^j \iff
\begin{aligned}
BER(\mathcal{M}_{\mathcal{P}}^i) &< BER(\mathcal{M}_{\mathcal{P}}^j) \quad \wedge \\
TS(\mathcal{M}_{\mathcal{P}}^i) &> TS(\mathcal{M}_{\mathcal{P}}^j) \quad \wedge \\
HD(\mathcal{M}_{\mathcal{P}}^i) &< HD(\mathcal{M}_{\mathcal{P}}^j)
\end{aligned}
\tag{1}
$$

We refer to $\mathcal{M}_{\mathcal{P}}^i$ as the positive sample (denoted $\mathcal{M}_{\mathcal{P}}^+$) and $\mathcal{M}_{\mathcal{P}}^j$ as the negative sample (denoted $\mathcal{M}_{\mathcal{P}}^-$) for the pair. Using this rule, we construct a set of preference triplets of the form $(\mathcal{P}, \mathcal{M}_{\mathcal{P}}^+, \mathcal{M}_{\mathcal{P}}^-)$, which constitutes our preference dataset for reinforcement learning with human feedback.

## 3.3  Mesh Generation Post-training

While our pre-trained model produces topologically valid meshes, two persistent challenges remain: (1) localized geometric imperfections in high-curvature regions, and (2) inconsistent face density distribution causing aesthetic artifacts. Although DeepMesh [23] adopts RLHF for mesh refinement, its reward function is primarily based on global mesh structure, making it insufficient for fine-grained control over local mesh quality. To address these limitations, we propose Masked Direct Preference Optimization (M-DPO)—a spatially aware extension of DPO) [24]. M-DPO introduces quality localization masks to guide learning toward problematic regions, enabling more targeted and effective mesh refinement.

**Quality-Aware Local Masking.** The goal of local masking is to differentiate high-quality regions of a mesh from those of lower quality. Given a triangular mesh $\mathcal{M}$, we assess each triangle face individually. A face is labeled as *good* if it satisfies the following two conditions: (1) it can be successfully merged into a quadrilateral, and (2) the resulting quad has a quality score above a predefined threshold. The quad quality is evaluated using a weighted combination of three metrics introduced in Section 3.2: Angle Quality, Aspect Ratio, and Adjacent Consistency. For each triangle face labeled as *good*, we assign a value of 1 to all corresponding token positions in the mesh sequence (typically 9 tokens per face). Conversely, faces that do not meet the criteria are considered *bad*, and their associated tokens are assigned a value of 0. We define the local masking function as $\phi$, such that $\phi(\mathcal{M}) \in \{0,1\}^{|\mathcal{M}|}$, where $|\mathcal{M}|$ denotes the length of the token sequence representing mesh $\mathcal{M}$.

**Masked Direct Preference Optimization.** Standard DPO tends to optimize global reward signals uniformly across the entire mesh sequence, which can lead to over-smoothed results and the loss of fine-grained geometric details. In contrast, our Masked Direct Preference Optimization (M-DPO) addresses this limitation by applying element-wise importance weighting guided by local quality masks, allowing the model to focus refinement specifically on low-quality regions. As illustrated in Figure 3, we designate the pretrained model from the first stage as the reference model, denoted as $G_{\text{ref}} := G_{\theta}^{\text{pre}}$, whose parameters are frozen during training. A trainable policy model $G_{\psi}$ is then initialized with the parameters of $G_{\theta}^{\text{pre}}$, and subsequently fine-tuned to better align with human preferences by encouraging it to generate outputs closer to the positive examples in our preference dataset. The objective of M-DPO is to maximize the likelihood of preferred (positive) samples over less-preferred (negative) ones, with a focus on quality-critical regions identified via local masks:

$$
\mathcal{L}_{\text{M-DPO}}(\pi_{\psi}; \pi_{\text{ref}}) = -\mathbb{E}_{(\mathcal{P}, \mathcal{M}_{\mathcal{P}}^+, \mathcal{M}_{\mathcal{P}}^-) \sim \mathcal{D}} \left[ \log \sigma \left( \beta \mathcal{L}^+(\mathcal{P}, \mathcal{M}_{\mathcal{P}}^+) - \beta \mathcal{L}^-(\mathcal{P}, \mathcal{M}_{\mathcal{P}}^-) \right) \right]
\tag{2}
$$

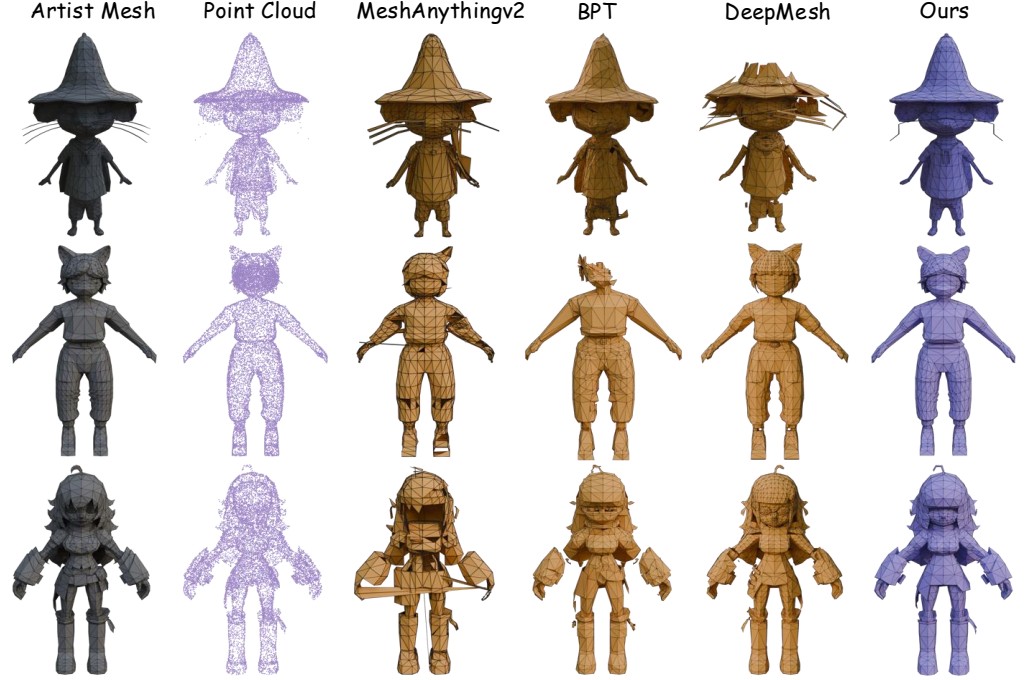

Figure 5: **Qualitative comparison on artist-designed meshes.** Our method generates more coherent and visually plausible surfaces with finer structural details and fewer topological artifacts compared to baseline approaches.

where the positive and negative log-ratio terms are computed as:

$$
\begin{aligned}
\mathcal{L}^+(\mathcal{P}, \mathcal{M}_{\mathcal{P}}^+) &= \log \frac{\|\pi_\psi(\mathcal{M}_{\mathcal{P}}^+|\mathcal{P}) \odot \phi(\mathcal{M}_{\mathcal{P}}^+)\|_1}{\|\pi_{\mathrm{ref}}(\mathcal{M}_{\mathcal{P}}^+|\mathcal{P}) \odot \phi(\mathcal{M}_{\mathcal{P}}^+)\|_1} \\
\mathcal{L}^-(\mathcal{P}, \mathcal{M}_{\mathcal{P}}^-) &= \log \frac{\|\pi_\psi(\mathcal{M}_{\mathcal{P}}^-|\mathcal{P}) \odot \left(1 - \phi(\mathcal{M}_{\mathcal{P}}^-)\right)\|_1}{\|\pi_{\mathrm{ref}}(\mathcal{M}_{\mathcal{P}}^-|\mathcal{P}) \odot \left(1 - \phi(\mathcal{M}_{\mathcal{P}}^-)\right)\|_1}
\end{aligned}
\tag{3}
$$

Here, $\mathcal{D}$ denotes the preference dataset, and $\pi$ is the token-level probability distribution produced by the model. The operator $\odot$ indicates element-wise (Hadamard) multiplication, and $\|\cdot\|_1$ denotes the $\ell_1$ norm over the token sequence. The hyperparameter $\beta$ controls the sharpness of preference separation, and $\sigma$ is the standard sigmoid function. M-DPO effectively preserves satisfactory regions while actively refining low-quality areas identified by the local quality mask. This targeted optimization strategy not only maintains the global structure but also enhances local geometric fidelity, offering a finer control over mesh generation quality compared to standard DPO.

## 4 Experiments

### 4.1 Experiment Settings

**Datasets** Our model is pretrained on 2M meshes from large-scale datasets including ShapeNetV2 [63], 3D-FUTURE [64], Objaverse [65], Objaverse-XL [66], and licensed assets. After filtering low-quality scans and poorly topologized CAD models, 800K meshes form the fine-tuning subset. For preference alignment, we construct a specialized dataset of 10,000 generated meshes, each paired with 8 topological variations derived from the same input point cloud. To enhance geometric generalization, meshes are perturbed at the vertex level and subsampled from an initial 50K-point cloud to 16,384 points, without enforcing watertightness. For evaluation, we employ two test sets: (1) 100 high-quality, artist-designed meshes for qualitative analysis, and (2) 100 dense, out-of-distribution meshes generated by Hunyuan2.5 [48], providing rigorous real-world validation. More data details can be seen in Supplementary A.1.

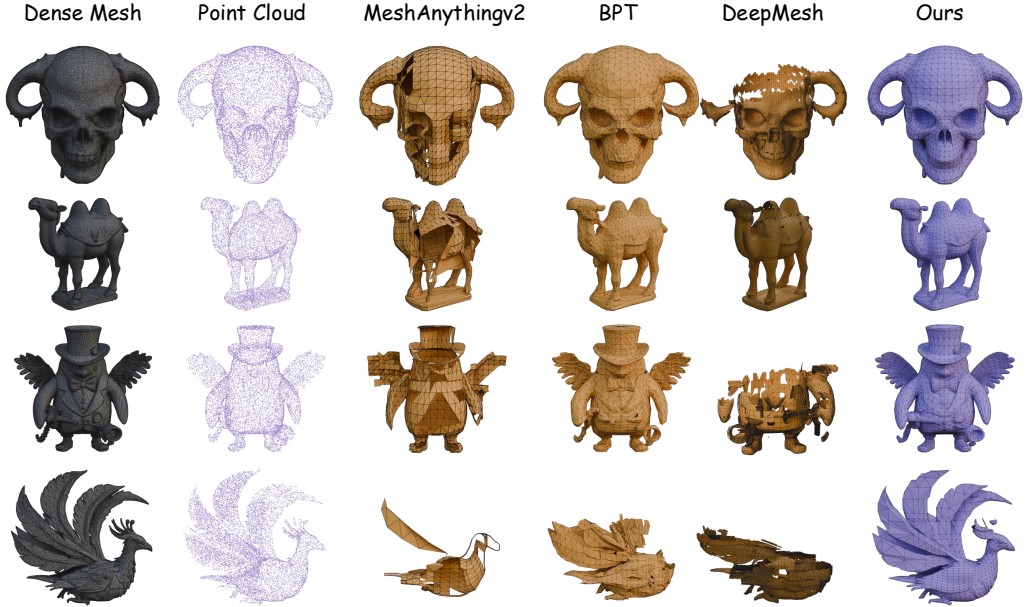

| Dense Mesh | Point Cloud | MeshAnythingv2 | BPT | DeepMesh | Ours |

Figure 6: **Generalization results on dense, out-of-distribution meshes.** Our model demonstrates superior geometric fidelity and surface continuity, maintaining high-quality reconstruction even under complex and unseen input conditions.

**Implementation Details**   We pretrained on 256 NVIDIA H20 GPUs (2/GPU) for 10 days with AdamW [67] ($\beta_1 = 0.9$, $\beta_2 = 0.99$) and Flash Attention, following a 100-step linear warm-up. M-DPO post-training took 8 hours on 64 GPUs with a $5e - 7$ learning rate. See supplementary material A.2 for full details.

**Baselines.**   We benchmark our approach against leading mesh generation methods, including **MeshAnythingV2** [16], **BPT** [19], and **DeepMesh** [23]. Since DeepMesh only publicly provides inference code and a 512M parameter version, we use this configuration for comparison.

## 4.2   Qualitative Results

We qualitatively compare our method with existing baselines. As shown in Figure 5, our model produces meshes that are significantly more coherent, artistically plausible, and faithful to the input geometry, particularly in challenging regions such as fine-grained structures and curved surfaces. These results highlight our model's ability to preserve detail and maintain topological regularity. In contrast, baseline methods often exhibit structural artifacts such as incomplete regions, broken connectivity, or excessive smoothing, especially in geometrically intricate areas. To further evaluate generalization beyond the training distribution, we conduct experiments on a set of dense, high-resolution meshes not seen during training. As illustrated in Figure 6, our method consistently outperforms prior approaches in reconstructing complex geometry and maintaining surface continuity under high-resolution inputs. These results demonstrate that our model not only performs well on curated artistic data but also generalizes effectively to challenging, real-world examples.

## 4.3   Quantitative Results

Table 1 presents a quantitative comparison of our method against baselines on artist-designed meshes and dense meshes derived from AI-generated representations. We report both geometric and topological metrics, including Hausdorff Distance (HD), Topology Score (TS), and Boundary Error Rate (BER). Our method consistently outperforms competing approaches across all metrics, demonstrating superior geometric fidelity and topological coherence. To further validate perceptual quality, we conducted a user study(US) in which participants were asked to compare mesh outputs based on visual plausibility and structural integrity. The results indicate a strong preference for our

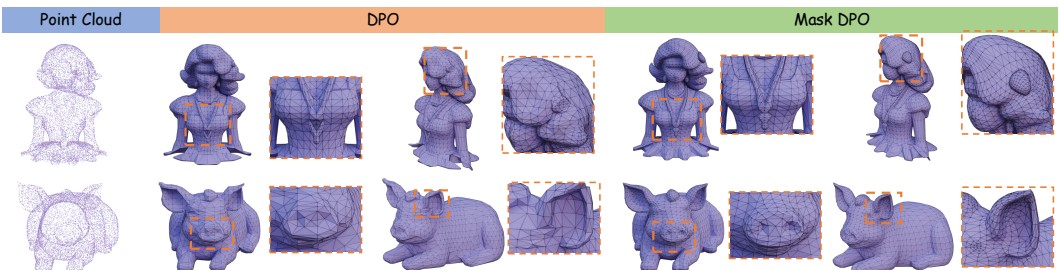

Figure 7: **The effectiveness of Mask DPO.** The addition of Mask DPO enhances the visual fidelity of the generated meshes, despite similar geometric performance across methods.

Table 1: **Quantitative comparison with other baselines in Artist and Dense Meshes.** Our approach achieves superior performance in both geometric accuracy and visual fidelity compared to existing baselines. DeepMesh* were tested using their 0.5 B version.

| Data Type | Artist Meshes | | | | | Dense Meshes | | | | |
|---|---|---|---|---|---|---|---|---|---|---|
| Metrics | CD ↓ | HD ↓ | TS ↑ | BER ↓ | US ↑ | CD ↓ | HD ↓ | TS ↑ | BER ↓ | US ↑ |
| MeshAnythingv2 [16] | 0.2143 | 0.4197 | 68.3 | 0.0749 | 9% | 0.2265 | 0.4760 | 72.0 | 0.0913 | 8% |
| BPT [19] | 0.1275 | 0.2735 | 72.7 | 0.0280 | 20% | 0.1615 | 0.3347 | 73.7 | 0.0113 | 18% |
| DeepMesh* [23] | 0.1331 | 0.2866 | 74.9 | 0.0296 | 22% | 0.1760 | 0.3570 | 75.8 | 0.0044 | 20% |
| **Ours** | **0.0973** | **0.1826** | **77.5** | **0.0182** | **45%** | **0.1286** | **0.2411** | **79.4** | **0.0015** | **40%** |

method, confirming that its advantages are not only quantitatively measurable but also perceptually significant.

## 4.4 Ablation Study

### 4.4.1 Score System

We evaluate the efficacy of our score-based preference system within the domain of dense mesh generation. As demonstrated in Table 2, employing only Hausdorff Distance to differentiate between high- and low-quality meshes (denoted as N-DPO) yields marginal improvements in geometric consistency over the pretrained model (Pretrain) and exhibits a decrease in the TS score. Conversely, leveraging our proposed composite scoring system (denoted as S-DPO) for the

Table 2: **Quantitative Evaluation of Score System and Mask DPO Methods.**

| Method | CD ↓ | HD ↓ | TS ↑ | BER ↓ | US ↑ |
|---|---|---|---|---|---|
| Pretrain | 0.1588 | 0.3196 | 76.5 | 0.0033 | 30% |
| N-DPO | 0.1455 | 0.2919 | 75.7 | 0.0028 | 32% |
| **S-DPO** | 0.1348 | 0.2625 | 77.9 | 0.0023 | 35% |
| **M-DPO** | **0.1286** | **0.2411** | **79.4** | **0.0015** | **40%** |

construction of preference data facilitates a substantial performance gain.

### 4.4.2 Mask DPO

Figure 4 illustrates that standard global DPO often fails to capture local variations in mesh quality. Our proposed topology-aware local mask mechanism effectively addresses this limitation by enabling the model to learn from spatially localized preference signals. Built on the preference dataset derived from our scoring system, the Mask-DPO model (denoted as M-DPO) demonstrates a clear advantage over the global score-based DPO baseline (S-DPO), as shown in Figure 7. This localized learning strategy leads to significant improvements in both quantitative metrics and human preference, as confirmed in Table 2. Notably, M-DPO produces outputs that are not only closer to the ground truth but also more consistently favored by human evaluators, providing strong empirical support for localized preference learning.

# 5 Conclusion

Generating high-quality 3D meshes remains a significant challenge. We introduced Mesh-RFT, a novel framework employing topology-aware scoring and Masked Direct Preference Optimization (M-DPO) for fine-grained refinement. By leveraging objective metrics and localized optimization, Mesh-RFT advances the state-of-the-art in automated mesh generation. Our approach significantly improves both the geometric accuracy and topological fidelity of generated meshes compared to previous methods. This work offers a substantial step forward in creating production-ready 3D assets for a wide range of applications. Limitations and future work are discussed in appendix C.

## Acknowledgements

This research was supported by fundings from the Hong Kong RGC General Research Fund (152244/21E, 152169/22E, 152228/23E, 162161/24E), Research Impact Fund (No. R5011-23F, No. R5060-19), Collaborative Research Fund (No. C1042-23GF), NSFC/RGC Collaborative Research Scheme (No. CRS_HKUST602/24), Areas of Excellence Scheme (No. AoE/E-601/22-R), and the InnoHK (HKGAI).

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

# Appendix

## A    More Implementation Details

### A.1    Data Details

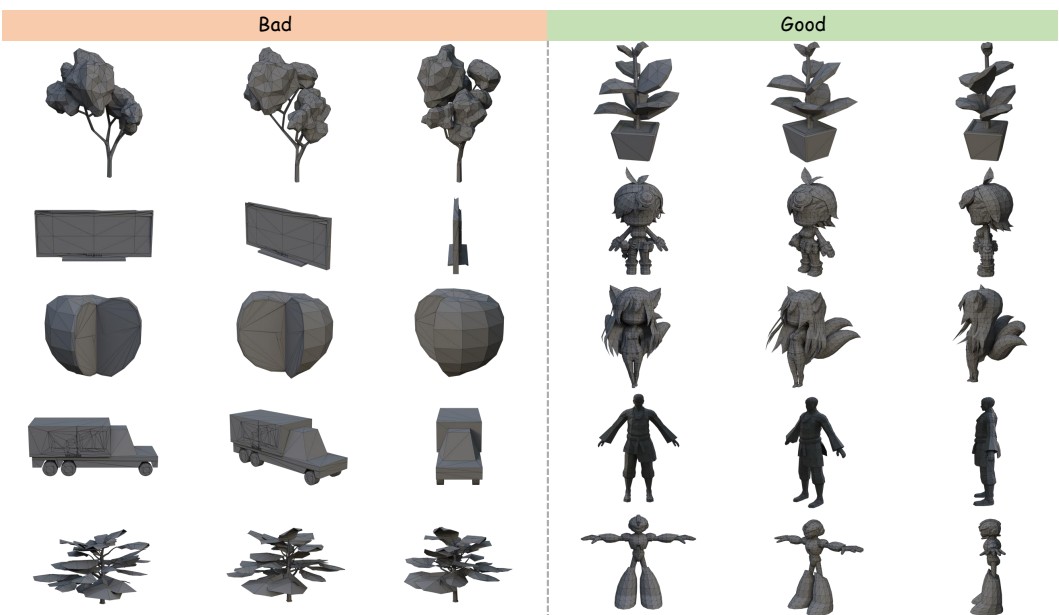

Figure 8: **Examples of Wiring Complexity in the Dataset.** The dataset contains cases with high-quality surface triangulations alongside instances where local regions exhibit lower quality.

After filtering low-quality scans and poorly topologized CAD models, our dataset size was reduced from 2 million to approximately 800,000 samples, with an average face count of 5,000. The distribution of face counts in this refined dataset is illustrated in Figure 9. Despite this initial filtering, as demonstrated in Figure 8, the dataset still includes instances where local surface triangulation quality is suboptimal. These instances are challenging to entirely eliminate due to the fact that even within lower-quality cases, regions with good topology often exist.

### A.2    More Training and Inference Details

Our model consists of 24 Transformer layers (1.1B parameters) arranged in a three-stage hourglass structure(2-4-12-4-2). It features a hidden dimension of 1536 and 16 attention heads. The vocabulary size for vertex coordinate quantization is 1024. The architecture supports a 36,864-token context window during inference and generates meshes through temperature-controlled sampling ($T = 0.5$), balancing output diversity and stability. For the pretraining phase, we initially trained on 2M meshes for 6 days, followed by an additional 4 days of training on a filtered set of 800k meshes. A 5k-face mesh from the preference dataset requires approximately 45k tokens. Generating 80,000 meshes from 10,000 dense meshes took about 2 days, with processing handled by 64 GPUs at a batch size of 8 per GPU, resulting in a speed of around 40 tokens/s.

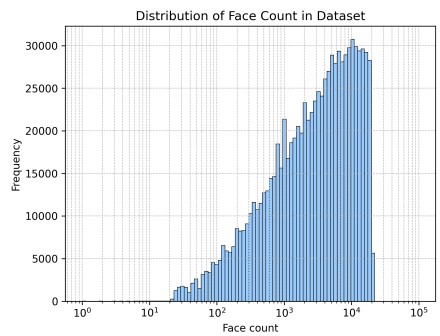

Figure 9: **Face Count Distribution in the Fine-tuning Dataset.** This figure presents the distribution of face counts within our fine-tuning dataset, which comprises approximately 800k samples with an average of 5k faces per model.

In contrast, calculating the EDR, TS, HD, and local mask for each mesh was completed in under 1 hour on a single machine. Furthermore, we utilize ZeRO-2 to minimize GPU memory consumption.

## A.3 Metrics Details

The Topology Score $TS(\mathcal{M})$ provides a quantitative measure of the structural quality of a mesh $\mathcal{M}$. It is computed based on the properties of a derived quadrilateral mesh $\mathcal{Q}(\mathcal{M})$ and is defined as a weighted linear combination of four sub-metrics:

$$TS(\mathcal{M}) = w_1 \cdot s_1(\mathcal{Q}(\mathcal{M})) + w_2 \cdot s_2(\mathcal{Q}(\mathcal{M})) + w_3 \cdot s_3(\mathcal{Q}(\mathcal{M})) + w_4 \cdot s_4(\mathcal{Q}(\mathcal{M})) \qquad (4)$$

where the weights are empirically set to $w_1 = 0.4$ (Quad Ratio), $w_2 = 0.2$ (Angle Quality), $w_3 = 0.3$ (Aspect Ratio), and $w_4 = 0.1$ (Adjacent Consistency), satisfying $\sum_{i=1}^{4} w_i = 1$. The sub-metrics are formally defined as follows:

- **Quad Ratio** ($s_1$): This metric assesses the efficiency of the triangle-to-quad conversion. Let $\mathcal{F}_{\mathcal{Q}}$ be the set of quadrilateral faces and $\mathcal{F}_{\mathcal{T}}$ be the set of triangular faces in $\mathcal{Q}(\mathcal{M})$. The Quad Ratio is given by:

$$s_1(\mathcal{Q}(\mathcal{M})) = \frac{|\mathcal{F}_{\mathcal{Q}}|}{|\mathcal{F}_{\mathcal{T}}| + |\mathcal{F}_{\mathcal{Q}}|} \qquad (5)$$

  where $|\cdot|$ denotes the cardinality of the set.

- **Angle Quality** ($s_2$): This metric quantifies the deviation of quadrilateral angles from the ideal $90°$. For each quadrilateral $q \in \mathcal{Q}(\mathcal{M})$, let $A(q) = \{\alpha_1^q, \alpha_2^q, \alpha_3^q, \alpha_4^q\}$ be the set of its internal angles. The Angle Quality is defined as the average normalized deviation:

$$s_2(\mathcal{Q}(\mathcal{M})) = 1 - \frac{1}{|\mathcal{Q}(\mathcal{M})|} \sum_{q \in \mathcal{Q}(\mathcal{M})} \frac{\sum_{\alpha \in A(q)} |\alpha - 90°|}{360°} \qquad (6)$$

- **Aspect Ratio** ($s_3$): This metric evaluates the regularity of the quadrilateral shapes. For a quadrilateral $q \in \mathcal{Q}(\mathcal{M})$ with side lengths $l_{q,1}, l_{q,2}, l_{q,3}, l_{q,4}$, the aspect ratio $r_q$ is defined as:

$$r_q = \max\left(\frac{\max(l_{q,1}, l_{q,3})}{\min(l_{q,1}, l_{q,3})}, \frac{\max(l_{q,2}, l_{q,4})}{\min(l_{q,2}, l_{q,4})}\right) \qquad (7)$$

  An additional edge ratio $e_q$ for each quadrilateral is computed as the average of its side lengths normalized by the maximum side length:

$$e_q = \frac{1}{4} \sum_{i=1}^{4} \frac{l_{q,i}}{\max_{j=1}^{4} l_{q,j}} \qquad (8)$$

  The Aspect Ratio sub-metric $s_3$ is then a combination of these measures:

$$s_3(\mathcal{Q}(\mathcal{M})) = 0.5 \cdot \left(\frac{1}{\frac{1}{|\mathcal{Q}(\mathcal{M})|} \sum_{q \in \mathcal{Q}(\mathcal{M})} r_q}\right) + 0.5 \cdot \left(\frac{1}{|\mathcal{Q}(\mathcal{M})|} \sum_{q \in \mathcal{Q}(\mathcal{M})} e_q\right) \qquad (9)$$

- **Adjacent Consistency** ($s_4$): This metric encourages smooth variations in the aspect ratios of neighboring quadrilaterals. For a quadrilateral $q_i \in \mathcal{Q}(\mathcal{M})$, let $\mathcal{N}(q_i)$ be the set of its adjacent quadrilaterals, and let $r_{q_j}$ be the aspect ratio of a neighboring quadrilateral $q_j \in \mathcal{N}(q_i)$ (calculated as in Equation 7). The average aspect ratio difference for $q_i$ is:

$$d_{q_i} = \frac{1}{|\mathcal{N}(q_i)|} \sum_{q_j \in \mathcal{N}(q_i)} |r_{q_i} - r_{q_j}| \qquad (10)$$

  The Adjacent Consistency sub-metric $s_4$ is then defined as the average of a consistency score based on this difference over all quadrilaterals:

$$s_4(\mathcal{Q}(\mathcal{M})) = \frac{1}{|\mathcal{Q}(\mathcal{M})|} \sum_{q \in \mathcal{Q}(\mathcal{M})} \frac{1}{1 + d_q} \qquad (11)$$

# B More Results

We present further comparative results in Figure 10 and Figure 11, respectively. **MeshAny-thingV2** [16], due to its Adjacent tokenizer, frequently exhibits line-shaped discontinuities. **BPT** [19], employing a block patch-based tokenizer, is prone to generating patch-level holes. **DeepMesh** [23] 512M version demonstrates significant instability. While exhibiting better topological visual quality, likely due to the use of truncated training and global-reward DPO, it generates excessively dense meshes lacking the adaptive tessellation characteristic of artist-designed meshes. Our method, which incorporates M-DPO, achieves superior visual quality and mesh tessellation.

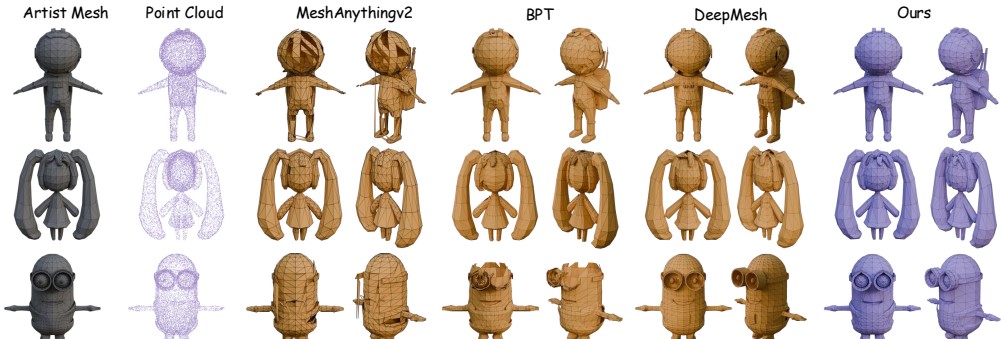

Figure 10: **Comparative Results for Mesh-RFT and Baseline Methods on Artist-Designed Meshes.**

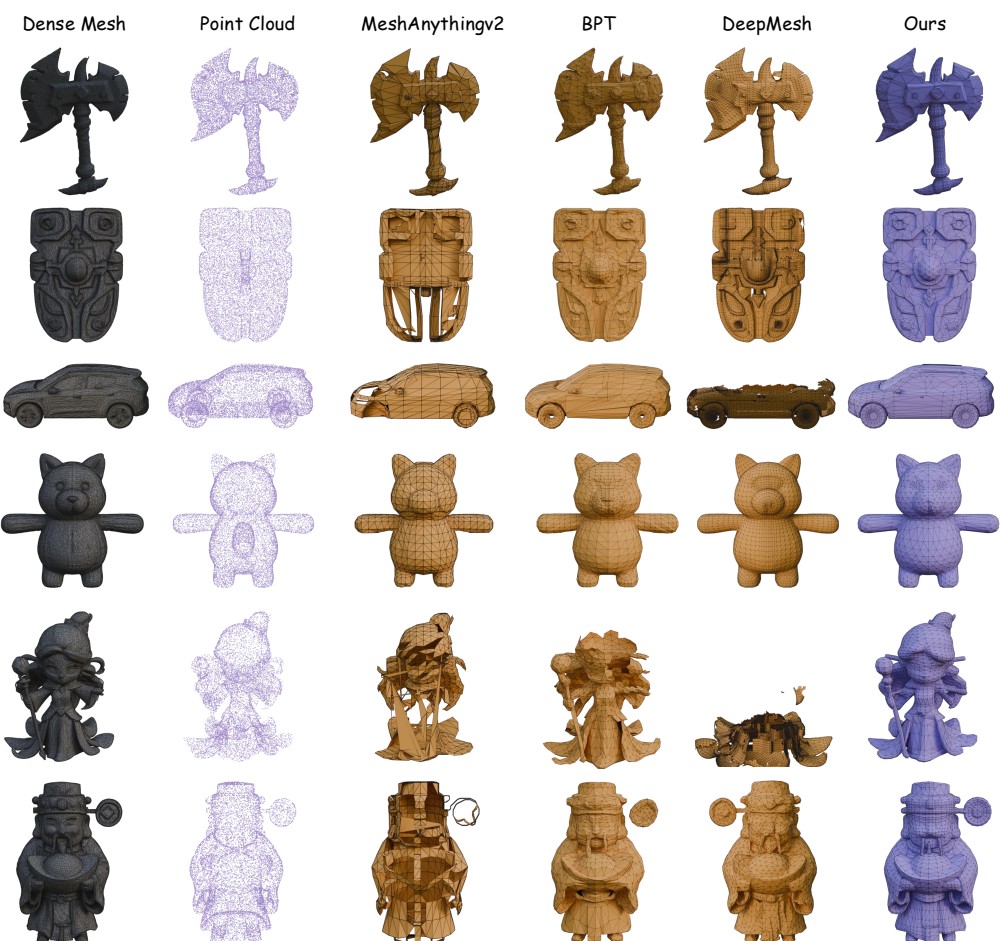

Figure 11: **Comparative Results for Mesh-RFT and Baseline Methods on AI-Generated Dense Meshes.**

# C   Limitations and Future Work

**Computational Efficiency**   While Mesh-RFT demonstrates significant advancements in mesh generation, its computational efficiency warrants further investigation. Exploring engineering optimizations, potentially drawing inspiration from efficient inference techniques such as vLLM [68] employed in large language models, could lead to substantial accelerations.

**Topological Correctness in Complex Geometries**   Ensuring robust topological correctness, particularly for intricate object geometries, necessitates continued research. As depicted in Figure 12, our model can exhibit topological defects such as holes in complex geometric scenarios. This may stem from limitations in the representational capacity of the point cloud encoder to capture fine-grained details within these complex structures. Future directions could involve leveraging more powerful, pre-trained point cloud encoders, increasing the number of tokens utilized, and scaling the decoder parameters to enhance the model's ability to discern intricate geometric features.

**Conditioning Modality**   As illustrated in Figure 12, dense meshes generated by Hunyuan2.0 [48] can sometimes exhibit a loss of fine details. Furthermore, conditioning on point clouds sampled from watertight dense meshes may exacerbate this information loss. Future work could explore alternative conditioning strategies, potentially bypassing the intermediate dense mesh representation and directly generating artist-quality meshes from image inputs (image-to-mesh generation).

**Topology Reward Refinement**   The current reward function is relatively basic. Future research could focus on exploring more generalized and sophisticated topology rewards, as well as integrating real-time, state-of-the-art reinforcement learning strategies [69, 70].

Addressing these limitations will be crucial for broadening the applicability and enhancing the robustness of Mesh-RFT across a wider range of diverse and challenging 3D modeling tasks.

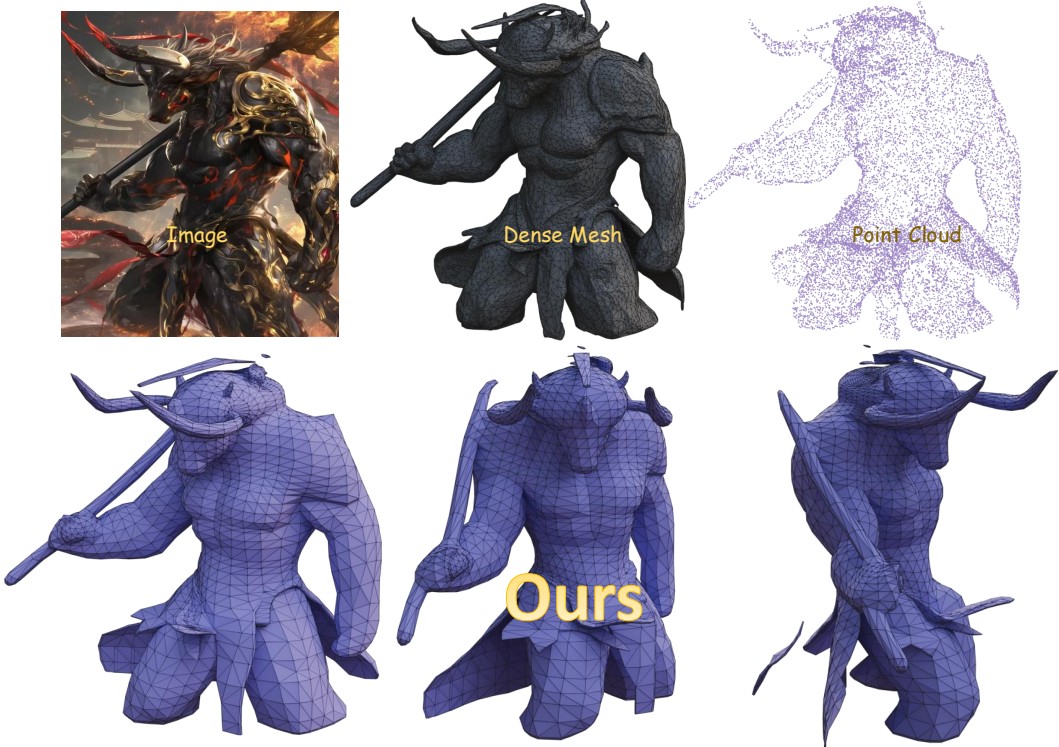

Figure 12: **Limitations of Mesh-RFT.** Examples showcasing potential topological defects (holes) in complex geometries and loss of fine details in generated meshes.

