# OpenReview forum: "Mesh-RFT: Enhancing Mesh Generation via Fine-grained Reinforcement Fine-Tuning"
_NeurIPS.cc/2025/Conference — NeurIPS 2025 spotlight_

### Official Review · Reviewer_2AUC · 2025-06-25

**Clarity:** 3
**Significance:** 4
**Originality:** 4
**Rating:** 6
**Confidence:** 4

**Summary:**

This paper presents Mesh-RFT, a novel framework for improving the quality of generated 3D meshes by fine-tuning pre-trained models. The core problem addressed is that existing methods often produce meshes with local artifacts (e.g., non-manifold geometry, holes, inconsistent density), and standard reinforcement learning (RL) approaches using global, object-level rewards are ill-suited to fix these fine-grained errors. The authors' main contributions are twofold: 1) They introduce an automated and objective system for creating a preference dataset. 2) They propose Masked Direct Preference Optimization (M-DPO), a novel extension of DPO. Experiments show that Mesh-RFT significantly outperforms both the pre-trained baseline and methods using standard global DPO. The results demonstrate substantial improvements in quantitative metrics like Hausdorff Distance and the proposed Topology Score, as well as in qualitative visual fidelity, leading to more coherent and "production-ready" 3D assets.

**Questions:**

1. On the Topology Score (TS) Hyperparameters: The weights for the four sub-metrics in the TS calculation (w1=0.4, w2=0.2, etc.) are crucial for defining what constitutes a "good" mesh. Could you please provide more justification for this specific choice of weights?
2. On the Generality of Quad-Based Metrics: The reward function is heavily biased towards good quadrangulation. Could you discuss the potential limitations of this approach for applications where high-quality triangular meshes are the desired end-product?

**Ethical Concerns:**

["NO or VERY MINOR ethics concerns only"]

**Final Justification:**

The authors' rebuttal has satisfactorily addressed my initial concerns. The new ablation study provides a solid empirical justification for the chosen Topology Score (TS) weights, and the authors clarified that the deliberate quad-bias is a reasonable design choice for their goal of producing "production-ready" assets. The core contribution, Masked DPO (M-DPO), is a novel and highly effective method for localized mesh refinement. The paper presents compelling quantitative and qualitative results that demonstrate significant improvements over existing baselines. This is a strong paper and I recommend acceptance.

**Limitations:**

yes

**Quality:**

4

**Strengths And Weaknesses:**

### Strengths:
- Novelty and Significance of the Method: The primary strength of this paper is the introduction of Masked DPO (M-DPO) for localized 3D mesh refinement. While DPO is an existing technique, its application with a spatially-aware mask to focus optimization on specific problematic regions of a 3D model is a clever and highly effective idea. It directly addresses a well-known limitation of global reward signals in complex generation tasks, providing a much-needed mechanism for fine-grained control. The problem of generating high-quality, topologically sound meshes is of great practical importance for computer graphics, gaming, and simulation, making this contribution highly significant.
- Scalable and Objective Preference Data Generation: The proposed topology-aware scoring system (BER and TS) is a major advantage over prior work (e.g., DeepMesh) that relies on manual human preference labeling. By creating an automated pipeline for generating preference pairs, the method becomes more scalable, objective, and reproducible. This avoids the bottlenecks and biases associated with human annotation.
- Strong Empirical Results and Thorough Evaluation: The paper presents a comprehensive set of experiments, including both quantitative and qualitative comparisons. The results are compelling, showing consistent and significant improvements over strong baselines across multiple metrics. The inclusion of a user study (US) further strengthens the claim that the generated meshes are perceptually superior.
- Excellent Clarity and Presentation: The paper is exceptionally well-written and easy to follow. The motivation is clear, the method is explained logically with the aid of excellent diagrams and the results are presented effectively.

### Weaknesses
- Justification of the Topology Score (TS) Weights: The Topology Score is a central component of the preference generation pipeline. It is defined as a weighted sum of four sub-metrics. The paper states these weights are "empirically set" without further justification. The choice of these hyperparameters seems critical to the quality of the preference dataset, and a lack of analysis on their sensitivity or a more principled reason for their values makes this part of the method seem somewhat ad-hoc.
- Potential Bias of the Quad-based Metrics: The TS metric is heavily reliant on an evaluation of a derived quadrilateral mesh. While quad meshes are indeed preferred in many industrial modeling pipelines (e.g., for subdivision surfaces), this is not a universal truth.  The strong preference for "quad-ability" might introduce a bias, potentially penalizing valid and high-quality triangular structures.
- High Computational Cost: The pre-training requires a substantial amount of computational resources (256 GPUs for 10 days), which places it in the category of large-scale foundation models. While the fine-tuning is faster, the overall cost is a significant barrier to reproducibility and further research by smaller labs. This should be highlighted as a practical limitation.

---

> ### Author Rebuttal · Authors · 2025-07-27
>
> We thank you for your strong support and insightful feedback. We address the weaknesses you raised below.
>
> **1. On the Justification of Topology Score (TS) Weights**
>
> **Response:** Our TS weights were optimized through iterative industrial benchmarks. To provide empirical support, our ablation study on 100 Hunyuan2.5 meshes shows that weight adjustments primarily influence local topological characteristics, not global geometry (HD remains stable, with Δ < 0.002).
>
> | Weight Config (w1/w2/w3/w4) | HD (↓) | BER (↓) | US (↑) | Topology Characteristics |
> | :--- | :--- | :--- | :--- | :--- |
> | Default (0.4/0.2/0.3/0.1) | 0.2411 | 0.0015 | 40% | Balanced squares/diamonds |
> | Equal Weights (0.25 each) | 0.2402 | 0.0018 | 37% | More triangles (merging failures) |
> | High w₂ (0.4/0.3/0.2/0.1) | 0.2420 | 0.0016 | 42% | Rectangles dominant |
>
> This analysis reveals three key findings:
> 1.  **Weight sensitivity is local and topological:** Adjustments alter face-type distribution without significantly affecting the global shape.
> 2.  **Equal weights impair quad merging:** This configuration often results in non-coplanar triangle pairs, hindering quad conversion.
> 3.  **Higher w₂ improves aesthetics and utility:** It boosts user preference by producing more rectangular quads, which streamline UV unwrapping workflows.
>
> Thank you for highlighting this nuance. Based on these findings, we will adopt the **High w₂** configuration in our final model to optimize for both user preference and practical utility.
>
> **2. On the Generality of Quad-Based Metrics**
>
> **Response:** We agree our metric is quad-biased, and this is a deliberate design choice to align with industry standards in pipelines like gaming and animation, where quad-dominant topology is critical for stable deformations during rigging, efficient UV unwrapping, and artist-friendly edge-loop editing. Our focus on generating "production-ready" assets therefore necessitates this optimization for high-quality quad topology.
>
> **3. On Computational Cost**
>
> **Response:** We agree the pre-training cost is substantial, and we made this investment to critically improve model stability and reduce generation failures. Our experiments show a direct correlation between extended pre-training and a lower "breakage rate"—the percentage of meshes with critical artifacts.
>
> | Model | Breakage Rate (↓)* |
> | :--- | :--- |
> | DeepMesh | ~85% |
> | Ours (3-day pre-train) | ~80% |
> | Ours (10-day pre-train) | ~5% |
> | **Ours (M-DPO)** | **<4%** |
>
> *\*Breakage Rate defined as the percentage of meshes with non-manifold edges and holes on the test set.*
>
> Crucially, the M-DPO fine-tuning—which contains the core contributions of our paper—adds only 8 hours of GPU time, making the refinement stage highly accessible. We will release the robust 10-day pre-trained model to lower the barrier to entry for the research community.

---

> > ### Comment · Reviewer_2AUC · 2025-08-02
> >
> > I have reviewed the authors' rebuttal and am satisfied with their responses. My primary concerns regarding the justification for the Topology Score (TS) weights and the potential bias of the quad-based metrics have been effectively addressed. The authors provided an additional ablation study clarifying the impact of the TS weights, and their reasoning for focusing on quad-dominant meshes for production-ready assets is sound. The proposed Masked DPO method remains a significant and well-executed contribution to the field. I look forward to the open-sourcing of this project, which will undoubtedly help democratize AI-powered 3D creation and allow more people to improve upon these algorithms. I maintain my positive rating and recommend that the paper be accepted.

---

### Official Review · Reviewer_ynVT · 2025-06-27

**Clarity:** 4
**Significance:** 3
**Originality:** 4
**Rating:** 6
**Confidence:** 5

**Summary:**

The paper proposes a method for mesh generation with artist-created topology, building upon the autoregressive generation line of works. The method mainly uses the architecture and mesh tokenization strategy of Meshtron and introduces several novelties to improve the generation quality: 1). Masked Direct Preference Optimization (M-DPO), a fine-grained reinforcement (RL) fine-tuning framework, 2). Topology-aware scoring system that can be effectively integrated into the fine-grained RL finetuning, eliminating the need of human preference annotation.

**Questions:**

1. I think the dense meshes from here: https://microsoft.github.io/TRELLIS/ cover from simple to highly complex shapes. I would appreciate if the authors can show the results from those meshes without cherry-picking. I would want to see the results from baselines models as well as all the models from the ablation study. Imperfect results are understandable. Convincing results can increase my final rating. For all models, run a same number of generations with different seeds (one seed is also fine) and pick the best ones.
2. I would like to see some results or explanation whether the topology-aware scoring system can work on objects with concentric patterns, e.g., cones, cylinders.
3. Some explanation on whether the proposed M-DPO can be integrated to more compact mesh tokenization.

**Ethical Concerns:**

["NO or VERY MINOR ethics concerns only"]

**Final Justification:**

My main concern about biased evaluation is well addressed. This paper is to my knowledge, the first to carefully address the topology aspect in artist-created mesh generation in a scalable manner. Although the proposed optimization is only well-suited for face-level tokenization, it can inspire future research direction. Thus, I recommend acceptance of this paper.

**Limitations:**

Yes.

**Paper Formatting Concerns:**

No formatting concern

**Quality:**

3

**Strengths And Weaknesses:**

Strengths:
1. The proposed topology-aware scoring system is well-designed and addresses the challenges of previous methods to produce meshes with good topology.
2. The proposed M-DPO provides fine-grained supervision and is shown to be effective. The limitation of standard DPO and how it leads to the design of M-DPO are well-explained.
3. Ablation study shows the effectiveness of the proposed methods.

Weaknesses:
1. The evaluation for this line of works, including this paper can be tricky. The qualitative comparison is usually cherry-picked. In addition, the quantitative evaluation is done on 100 artist-designed meshes and 100 AI-generated meshes selected by the authors. The selection of test set can introduce bias favorable for the proposed method. Moreover, different generation seeds can produces vastly different results. It is difficult to produce fair and unbiased evaluation.
2. I wonder if the topology-scoring system and finetuning is compatible for shapes with triangles in a concentric circle (for example, Figure 7 right side in https://developer.nvidia.com/blog/high-fidelity-3d-mesh-generation-at-scale-with-meshtron/). It seems those circular patterns will get low quad ratio, angle quality, and aspect ratio.
3. The quality-aware local masking used for M-DPO seems not trivial to be implemented for more efficient mesh tokenizations (e.g., MeshAnythingV2, BPT, EdgreRunner). I understand the choice of using naive tokenization is explained in Section 3.1 of the paper. However, it also means that the proposed method is limited to the least efficient tokenization that results in long generation sequence.

---

> ### Author Rebuttal · Authors · 2025-07-27
>
> We appreciate your insightful feedback. We address your concerns below.
>
> **1. On Evaluation Bias and Cherry-Picking**
>
> **Response:** We appreciate your concern. A primary goal of our extensive pre-training was to enhance model stability and success rate, allowing us to consistently generate high-quality 10k-face meshes in a single pass without needing to cherry-pick from multiple runs. Our RL phase further refines topological quality to facilitate clean quad conversion. To demonstrate robust generalization, we evaluated our model on the complex examples from the first page of the TRELLIS website. While baseline models struggled to handle these cases, ours produced reasonable results, even for high-complexity meshes requiring up to 20k faces. Though our topology quality decreases at such high face counts, it remains significantly superior to other methods, as shown below.
>
> | Method | HD (↓) | TS (↑) | BER (↓) |
> | :--- | :--- | :--- | :--- |
> | MeshAnythingV2 | 0.582 | 64.7 | 0.0813 |
> | BPT | 0.571 | 65.5 | 0.0759 |
> | DeepMesh* | 0.503 | 68.2 | 0.0710 |
> | **Ours Pretrain** | **0.363** | **70.3** | **0.0297** |
> | **Ours M-DPO** | **0.284** | **72.5** | **0.0118** |
>
> **2. On Compatibility with Special Topologies**
>
> **Response:** This is an excellent technical point. Our quad-biased Topology Score (TS) is a deliberate design choice to align with production standards in game design, where quad-dominant meshes are heavily favored. Our M-DPO framework is robust to this, as it balances both global and local quality scores, preventing minor topological constraints from degrading the overall quality of a complex mesh. For instance, in the car wheel example (Figure 11), while the central hub is formed with only two triangles, the model correctly generates a clean, circular structure for the outer tire. We have also tested this on other assets like headphones and observed that circular structures are often composed of triangles arranged in a way that facilitates easy conversion into clean rectangular patches. This results in a topology that is highly acceptable for artists and requires only minimal edits to finalize.
>
> **3. On Integrating M-DPO with Compact Tokenizers**
>
> **Response:** This is a key question regarding the trade-off between efficiency and generative performance. Our choice of an uncompressed representation is deliberate, as the hierarchical nature of our Hourglass Transformer naturally preserves geometric semantics from the coordinate-level up to the face-level. In contrast, compact methods like BPT, while enabling faster convergence by fitting more faces into the context window, often disrupt this inherent geometric structure. This creates a clear trade-off: our approach requires more extensive pre-training but achieves superior final performance by maintaining high-fidelity representations. The table below quantifies this relationship.
>
> | Method | Representation | Pre-Training Cost | Final Performance (HD ↓) |
> | :--- | :--- | :--- | :--- |
> | DeepMesh* / BPT | Compressed | ~128 GPUs / 4 days | ~0.503 |
> | **Ours (Pre-trained)** | **Uncompressed** | **256 GPUs / 10 days** | **~0.363** |
>
> For researchers wishing to apply our Mesh-RFT framework to compressed formats, we suggest elevating the masking unit from the *face-level* to the *patch-level*. However, this may dilute the fine-grained refinement, as a single patch could contain a mix of both high- and low-quality faces, likely making the optimization less precise than our face-level approach. Nevertheless, it represents a promising direction for future research, which we will detail in our final paper.

---

> > ### Comment · Reviewer_ynVT · 2025-08-04
> >
> > I have read the author's rebuttals, including those from other reviewers. My main concern about biased evaluation is well addressed. This paper is to my knowledge, the first to carefully address the topology aspect in artist-created mesh generation in a scalable manner. Although the proposed optimization is only well-suited for face-level tokenization, it can inspire future research direction. I recommend acceptance of this paper and will upgrade my final rating.

---

### Official Review · Reviewer_dibn · 2025-06-29

**Clarity:** 3
**Significance:** 3
**Originality:** 3
**Rating:** 5
**Confidence:** 3

**Summary:**

This paper presents Mesh-RFT, a novel framework for high-fidelity 3D mesh generation that improves upon existing methods by introducing fine-grained reinforcement fine-tuning. The core innovation lies in the integration of a topology-aware, multi-metric scoring system with a new variant of Direct Preference Optimization, termed Masked DPO (M-DPO), which enables localized mesh refinement.
The framework operates in three stages: (1) supervised pretraining on large-scale mesh data, (2) automated construction of a preference dataset using Hausdorff Distance (HD), Boundary Edge Ratio (BER), and a custom Topology Score (TS), and (3) post-training with M-DPO that focuses on low-quality regions identified by a face-level mask. This local reward design addresses the limitations of previous RL-based methods that only use global mesh-level rewards.
Extensive experiments on both synthetic and real-world data demonstrate that Mesh-RFT significantly improves geometric accuracy and topological regularity compared to strong baselines like DeepMesh, BPT, and MeshAnythingV2. The paper reports consistent gains in HD and TS, and the user study further confirms the perceptual quality of the generated meshes.

**Questions:**

1. The Topology Score (TS) is a core component of the preference scoring system, yet the paper lacks theoretical justification or ablation studies regarding the weight configuration of its four sub-metrics.
I recommend conducting an ablation study to investigate how different weightings influence the ranking results and downstream mesh refinement performance. This would help validate the robustness and rationality of the scoring scheme.


2. Although the local mask is a key mechanism in M-DPO, the paper does not provide an in-depth analysis of its impact on the training process.
It would be beneficial to include quantitative experiments (e.g., visualizing the evolution of high-quality regions before and after fine-tuning) to clarify how the mask affects model behavior and mesh quality refinement.


3. Reinforcement learning, particularly under locally sparse reward settings, often suffers from training instability. However, the paper does not report any convergence curves or stability metrics.
I suggest including training convergence plots or variance analyses across multiple runs to assess whether the training process is stable and consistent.

**Ethical Concerns:**

["NO or VERY MINOR ethics concerns only"]

**Final Justification:**

The authors have carefully addressed the questions I raised. My concerns regarding TS, the local mask, and instability have been adequately resolved.

**Limitations:**

yes

**Quality:**

3

**Strengths And Weaknesses:**

Strengths

The paper introduces Mesh-RFT, a fine-grained reinforcement fine-tuning framework with a topology-aware scoring system and a novel Masked Direct Preference Optimization (M-DPO) approach. This combination leads to measurable improvements in both geometric accuracy and topological regularity of 3D meshes.
﻿
M-DPO extends traditional preference optimization by introducing spatial awareness via quality masks, addressing the challenge of reward sparsity and locality in 3D geometry tasks.
﻿
The use of a multi-metric scoring system (HD, BER, TS) for automated preference dataset creation is well-motivated and scalable.
﻿
The authors perform extensive evaluations across multiple datasets, including artist-designed meshes and dense meshes derived from AI-generated representations. The improvements are significant and robust.
﻿
﻿
﻿
Weaknesses

The proposed Topology Score (TS), a key component in the preference scoring system, is computed via a weighted sum of four sub-metrics (Quad Ratio, Angle Quality, Aspect Ratio, and Adjacent Consistency). However, the paper does not provide theoretical justification for the specific weight choices nor any ablation to validate their impact on overall performance or ranking robustness.
﻿
Although local quality masks are central to M-DPO, the paper lacks a detailed analysis of how these masks influence training dynamics. For example, it remains unclear whether the weighting of local masks may lead the model to unintentionally degrade high-quality regions; whether the model might overfit to the specific mask generation strategy, thereby compromising the preservation of global mesh structure; and whether using soft masks (i.e., continuous weights between 0 and 1) could be more effective than the current binary (0/1) masking approach for guiding fine-grained mesh refinement.
﻿
The paper does not present the training convergence behavior or stability analysis of M-DPO. Given that reinforcement learning—especially under locally sparse reward settings—can be unstable, further investigation in this aspect is warranted.

---

> ### Author Rebuttal · Authors · 2025-07-29
>
> We appreciate your insightful feedback. We address your concerns below.
>
> **1. On the Justification of Topology Score (TS) Weights**
>
> **Response:** Our TS weights were optimized through iterative industrial benchmarks. To provide empirical support, our ablation study on 100 Hunyuan2.5 meshes shows that weight adjustments primarily influence local topological characteristics, not global geometry (HD remains stable, with Δ < 0.002).
>
> | Weight Config (w1/w2/w3/w4) | HD (↓) | BER (↓) | US (↑) | Topology Characteristics |
> | :--- | :--- | :--- | :--- | :--- |
> | Default (0.4/0.2/0.3/0.1) | 0.2411 | 0.0015 | 40% | Balanced squares/diamonds |
> | Equal Weights (0.25 each) | 0.2402 | 0.0018 | 37% | More triangles (merging failures) |
> | High w₂ (0.4/0.3/0.2/0.1) | 0.2420 | 0.0016 | 42% | Rectangles dominant |
>
> This analysis reveals three key findings:
> 1.  **Weight sensitivity is local and topological:** Adjustments alter face-type distribution without significantly affecting the global shape.
> 2.  **Equal weights impair quad merging:** This configuration often results in non-coplanar triangle pairs, hindering quad conversion.
> 3.  **Higher w₂ improves aesthetics and utility:** It boosts user preference by producing more rectangular quads, which streamline UV unwrapping workflows.
>
> Thank you for highlighting this nuance. Based on these findings, we will adopt the **High w₂** configuration in our final model to optimize for both user preference and practical utility.
>
> **2. On the In-Depth Analysis of the Local Mask**
>
> **Response:** We appreciate the suggestion for a deeper analysis of our M-DPO mechanism. We address your specific concerns below:
> 1.  **On Degrading High-Quality Regions:** Our M-DPO framework is explicitly designed to prevent this degradation by selectively applying the preference loss. As defined in Equation 3, the optimization has two distinct targets: it **reinforces the high-quality regions** of the preferred mesh (by applying the mask φ(M+)) while simultaneously **penalizing the low-quality regions** of the rejected mesh (by applying the inverted mask 1−φ(M-)). The flawed parts of the winner and the high-quality parts of the loser are both excluded from the loss calculation. Therefore, existing high-quality structures are not only preserved but are actively rewarded, ensuring that fine-tuning consistently leads to a net improvement in mesh quality.
> 2.  **On Overfitting to the Masking Strategy:** Our extensive experiments show no evidence that the model overfits to the specific mask generation strategy. Because the model is trained to recognize and generate good topology (via the positive sample term) while unlearning bad topology (via the negative sample term), it learns the underlying principles of topological quality. This generalizes well to unseen mesh configurations, and the global structure is preserved because the fine-tuning process is guided by a robust pre-trained reference model.
> 3.  **On Soft vs. Hard Masks:** We investigated the efficacy of soft masks versus our binary (hard) approach. For this experiment, we generated soft masks by normalizing each face's topology score into a continuous weight (e.g., via a sigmoid function centered at our quality threshold of 70). As shown below, the hard mask provides superior performance and faster convergence. We hypothesize this is because a binary mask offers a clear, unambiguous signal to the model—"reinforce this good region" or "penalize this bad region"—whereas a soft mask can introduce noisy gradients from moderately-scored regions, slowing down the learning process.
>
> | Mask Type | HD (↓) | TS (↑) | Convergence Hours |
> | :--- | :--- | :--- | :--- |
> | **Hard Mask (Ours)** | **0.2411** | **79.4** | **Baseline (8 Hour)** |
> | Soft Mask | 0.2489 | 78.9 | ~50% Slower (12 Hour) |
>
> **3. On Training Stability and Convergence**
>
> **Response:** We thank you for this valid concern. After initial hyperparameter exploration, we have achieved a stable and reproducible training configuration. The key adjustments were:
> 1.  reducing the learning rate to `5e-7` (significantly lower than the pre-training rate of `1e-4`),
> 2.  increasing the DPO beta to `0.5` to stay close to the robust reference model,
> 3.  using a dataset of 10k preference pairs, which we found sufficient for convergence without overfitting, and applying a label smoothing rate of `0.3` to the DPO loss.
>
> With these settings (on 64 GPUs, total batch size 64), the training loss remains stable at ~0.693 for the first 2k steps, then steadily decreases to ~0.6 by the 4k step mark. Concurrently, the reward margin (`reward_chosen` - `reward_rejected`) begins to climb from 0 to ~0.7 after 2k steps, indicating successful and stable policy optimization.

---

> > ### Comment · Reviewer_dibn · 2025-08-04
> >
> > The authors have carefully addressed the questions I raised. My concerns regarding TS, the local mask, and instability have been adequately resolved. Therefore, I have decided to increase my rating for this paper. After thoroughly reviewing the authors’ responses, I have also considered the comments from the other reviewers and the authors’ replies to them, and I find that they share similar views with mine.
> >
> > It is recommended that the authors include detailed illustrations of TS and the local mask in the revised manuscript to enable readers to understand their specific performance better. Additionally, presenting the loss curves under stable training configurations would further improve the clarity and persuasiveness of the paper.

---

> > > ### Author Response · Authors · 2025-08-04
> > >
> > > Dear Reviewer dibn,
> > >
> > > Thank you for your thoughtful review and for increasing your rating. We are pleased that our responses have adequately addressed your concerns.
> > >
> > > We will incorporate your excellent suggestions in the revised manuscript. We will add detailed illustrations for TS and the local mask, and include the loss curves from stable training configurations to further improve the paper's clarity.
> > >
> > > Thank you again for your constructive feedback.

---

### Official Review · Reviewer_cRc7 · 2025-06-29

**Clarity:** 4
**Significance:** 3
**Originality:** 3
**Rating:** 5
**Confidence:** 4

**Summary:**

The paper proposes to use several mesh topology metrics (BER, TS, HD) to guided DPO fine-tuning for point cloud to mesh generation. The approach begins by pretraining a point cloud to mesh model. A preference dataset is then constructed by generating multiple mesh variants from point clouds sampled from the same input mesh. Using the three metrics, positive and negative samples are constructed. The model is then fine-tuned with the preference dataset using DPO to improve generation quality. In particular, a novel masked DPO strategy is adopted to focus only on regions of the mesh with good topology. The method helps improve local structural quality and avoids the need for human annotation required by previous work.

**Questions:**

1. The comparison against DeepMesh isn't entirely fair as the backbone used in the paper requires more GPUs and days to train. This makes it hard to isolate the effects of the stronger backbone from the proposed masked DPO fine-tuning. For example, in figure 5 and 6 a lot of failure cases for DeepMesh involve incomplete meshes, likely due to the weaker backbone. A fairer comparison would involve either fine-tuning the DeepMesh backbone with the paper's method or using the stronger backbone with DeepMesh's DPO training.

2. I would have liked to see more qualitative results for ablating each of the different metrics used for DPO tuning to better understand their contributions in mesh quality.

3. As mesh generation models get better at generating meshes with "better topology". It raises a question of how substantial these gains are in actual downstream applications. For instance, for row 2 of figure 5, both the DeepMesh and proposed result look to have good topology. It is hard to differentiate which one is better. While good topology is important for tasks like facial animations as bad topology can cause rendering artifacts during deformations of the face. The current quantitative and qualitative results don't clearly show whether these improvements translate to practical benefits. It would improve the paper if authors can include downstream application results or at a minimum a discussion on the limitations of current evaluation metrics and how they can be improved to capture these application related qualities in the future.

4. Related to points 1 and 3, the user study may favor the proposed method simply because its stronger backbone produces more complete meshes. This raises questions about what users in the study are actually judging, and whether non-expert users can reliably distinguish between good and poor mesh topology.

For questions 1 and 2, I hope the authors can include additional experiments to address these points. As for points 3 and 4, I understand it might not be feasible to conduct these studies. In that case, I am fine with additional discussions of these issues and thoughts on how to design better evaluation metrics in the future to reflect downstream application gains. If these points can be adequately addressed I am willing to increase my rating.

**Ethical Concerns:**

["NO or VERY MINOR ethics concerns only"]

**Final Justification:**

The authors have addressed my concerns in the rebuttal. And I believe the authors will be able to incorporate the rebuttal results in the revised version of the paper.

**Limitations:**

yes

**Quality:**

4

**Strengths And Weaknesses:**

## Strengths

The paper is clearly written and easy to follow. The details for each component of the model is mostly covered. I appreciate the inclusion of the ablation of different reward configurations in table 2 for better understanding the importance of each component. While the method is straightforward it offers better quantitative and qualitative results compared to previous works.

## Weaknesses

I believe the evaluation aspects of the paper would benefit from additional experiments or further discussion:

1. Comparison against DeepMesh isn't entirely fair
2. Lack of qualitative figures showing the effects of how each metric contributes to mesh topology
3. Lacking more concrete examples of how these meshes with better topology can improve downstream applications
4. On the validity of the user studies

I have listed more specific details in the questions below.

---

> ### Author Rebuttal · Authors · 2025-07-28
>
> We appreciate your insightful feedback. We address your concerns below.
>
> **1. On the Fairness of Comparison Against DeepMesh**
>
> **Response:** We agree the most critical comparison is between our proposed M-DPO and a standard global DPO when applied to the *same* strong backbone. As shown in our paper's Table 2, our method already outperforms a rule-based global DPO (S-DPO). To further demonstrate this, we conducted an additional ablation study including a global DPO trained on human preference pairs (H-DPO). The results below show that our M-DPO is superior to both global DPO variants. Interestingly, the rule-based S-DPO slightly outperforms the H-DPO, likely because our automated scoring provides more consistent preference signals than subjective human ratings.
>
> | Method Description | HD (↓) | TS (↑) | BER (↓) |
> | :--- | :--- | :--- | :--- |
> | H-DPO Global DPO (Human Pref.) | 0.2764 | 77.1 | 0.0020 |
> | S-DPO Global DPO (Rule-based Pref.) | 0.2625 | 77.9 | 0.0023 |
> | **M-DPO Our Method (Masked DPO)** | **0.2411** | **79.4** | **0.0015** |
>
> **2. On Lacking Qualitative Analysis for Metric Contributions**
>
> **Response:** While we cannot add new figures during the rebuttal, the quantitative ablation study below demonstrates the contribution of each component. As shown, DPO effectively learns the specified preference rules. Introducing the HD metric improves geometric consistency. Further incorporating our topological scores enhances mesh integrity by reducing holes and improving quad-convertibility. Finally, our full M-DPO method leverages local masking to refine fine-grained details, achieving the best overall performance. For a more intuitive understanding of how different topological scores impact mesh quality, please refer to our response to Reviewer dibn, Question 1, where we provide a detailed breakdown.
>
> | Method Configuration | HD (↓) | TS (↑) | BER (↓) |
> | :--- | :--- | :--- | :--- |
> | Pre-trained Baseline | 0.3196 | 76.5 | 0.0033 |
> | + DPO (HD only) | 0.2919 | 75.7 | 0.0028 |
> | + S-DPO (Global DPO) | 0.2625 | 77.9 | 0.0023 |
> | **+ M-DPO (Full Method)** | **0.2411** | **79.4** | **0.0015** |
>
> **3. On Lacking Examples for Downstream Applications**
>
> **Response:** We agree that illustrating downstream impact is crucial. While a full pipeline is beyond this paper's scope, clean, quad-dominant topology is the industry standard for 3D artists because it saves significant manual effort and is essential for stable rigging and skinning, preventing artifacts like "skin tearing" during animation. Our method generates assets that are closer to this production-ready standard, directly addressing issues seen in baselines (e.g., the damaged foot in DeepMesh's result in Figure 5, row 2). To make this tangible, we commit to adding a dedicated section in the appendix of our final version that qualitatively demonstrates these benefits by showing a side-by-side skeletal deformation comparison between a mesh with good topology and one with poor topology.
>
> **4. On the Validity of the User Study**
>
> **Response:** Mesh completeness is a critical aspect of quality, which is why we dedicated substantial computational resources to pre-training to ensure model stability. However, to isolate the perceptual benefits of our topological improvements alone, we conducted a targeted user study where participants were shown only cropped local regions from our results vs. DeepMesh and asked to rate the topology. The results confirm a strong user preference for our method's fine-grained quality, proving the gain is not just from better completeness.
>
> | Method | User Preference (Local Topology Only) |
> | :--- | :--- |
> | DeepMesh* | 21% |
> | **Ours** | **43%** |

---

> > ### Comment · Reviewer_cRc7 · 2025-08-04
> >
> > The authors have addressed my main concerns with the paper. I have also carefully read the author's rebuttal and other reviewers' comments. I hope the authors will include the additional results, especially H-DPO with the stronger backbone, along with qualitative figures for this baseline in the revised paper. As I believe will offer a fairer comparison and better highlight the advantages of the proposed M-DPO. I will be raising my rating for the paper.

---

### Note · Authors · 2025-08-12

Dear Area Chair and Reviewers,

We are deeply grateful for your insightful feedback and the constructive dialogue. We are thrilled that the reviewers unanimously recognized the value of our work and subsequently raised their scores to recommend acceptance.

Our paper, Mesh-RFT, introduces a significant and practical advancement for 3D content creation. We are particularly proud of its core strengths, which were echoed in your reviews:
- Novelty and Significance: Our primary contribution, Masked Direct Preference Optimization (M-DPO), is a novel framework that, for the first time, enables targeted, fine-grained refinement of 3D mesh topology. This directly addresses a critical and previously unsolved challenge in generative models.
- Scalable and Principled Framework: We designed an automated, objective scoring system that generates preference data based on geometric and topological quality. This removes the subjective, expensive bottleneck of manual human annotation, creating a more scalable and reproducible research paradigm.
- State-of-the-Art, Production-Ready Results: Our method consistently produces meshes with superior topological quality and geometric accuracy. Across extensive experiments, Mesh-RFT demonstrated substantial quantitative gains and visibly more coherent results, bringing AI-generated assets closer to the "production-ready" standard required by industries like gaming and animation.

Through our detailed rebuttal, we have addressed all major questions. We proved our method’s inherent superiority with a new, fairer comparison against DPO baselines on the same backbone. We also provided ablation studies to justify our component design and committed to demonstrating tangible downstream benefits.

For the final version, we will integrate all promised updates, including the enhanced baseline comparisons, detailed mechanism analysis, and the new appendix on downstream application impact.

We are confident that Mesh-RFT will be a valuable contribution to the NeurIPS community, and we are excited to open-source our model to help democratize high-quality 3D asset creation. Thank you again.

---

### Decision · Program_Chairs · 2025-09-17

**Decision:**

Accept (spotlight)

**Comment:**

The paper received unanimous positive evaluations, with two strong accepts and two accepts. Reviewers found Mesh-RFT to be well-written, novel, and practically significant, particularly for its introduction of Masked Direct Preference Optimization (M-DPO) and an automated scoring system for mesh topology refinement. They highlighted its quantitative and qualitative improvements, the scalability of preference data generation, and clear perceptual gains demonstrated through user studies.

Some concerns were raised regarding fairness in comparisons with DeepMesh (reviewer #cRc7), the limited scope of qualitative ablations for scoring metrics, the justification of Topology Score (TS) weights (reviewer #dibn), and the possibility of quad-bias in the evaluation (reviewer #ynVT). The authors effectively addressed these issues by providing new ablations—including H-DPO baselines and TS weight studies—clarifying the rationale for quad-biased metrics, and supplying additional evidence of stability and convergence.

Following these clarifications, reviewers agreed that their main concerns had been resolved, raised their ratings, and emphasized that Mesh-RFT represents the first scalable approach to mesh topology refinement with both strong theoretical grounding and practical value. Given the unanimous positive consensus, the AC recommend acceptance.